# MAKE IT SING: ANALYZING SEMANTIC INVARIANTS IN CLASSIFIERS

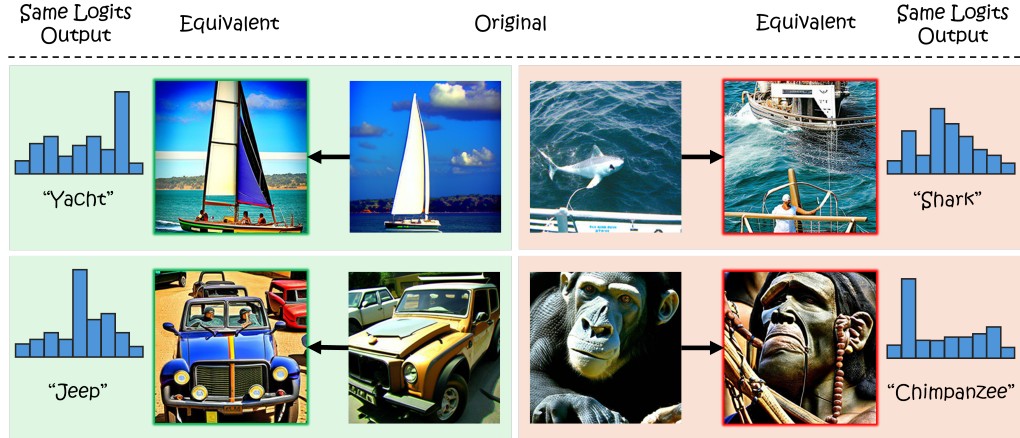

Figure 1: **Visualization of benign and problematic invariants.** The four images at the center correspond to certain features taken from a pretrained ResNet50. On the left and right columns their equivalent images are shown, following null-space removal. Each pair yields the same logits after passing through the linear head. The left side (green) demonstrates robustness, with little semantic change. The right side (red) incurs large semantic deviations. Our framework quantifies these changes statistically, diagnosing semantic invariants at the class and network level.

## ABSTRACT

All classifiers, including state-of-the-art vision models, possess invariants, partially rooted in the geometry of their linear mappings. These invariants, which reside in the null-space of the classifier, induce equivalent sets of inputs that map to identical outputs. The semantic content of these invariants remains vague, as existing approaches struggle to provide human-interpretable information. To address this gap, we present *Semantic Interpretation of the Null-space Geometry* (SING), a method that constructs equivalent images, with respect to the network, and assigns semantic interpretations to the available variations. We use a mapping from network features to multi-modal vision language models. This allows us to obtain natural language descriptions and visual examples of the induced semantic shifts. SING can be applied to a single image, uncovering local invariants, or to sets of images, allowing a breadth of statistical analysis at the class and model levels. For example, our method reveals that ResNet50 leaks relevant semantic attributes to the null space, whereas DINO-ViT, a ViT pretrained with self-supervised DINO, is superior in maintaining class semantics across the invariant space.

## 1 INTRODUCTION

State of the art networks, especially vision classifiers, learn internal representations with complex geometry; while this correlates with strong performance on recognition benchmarks, it makes mechanistic interpretability difficult (Doshi-Velez & Kim, 2017; Ansuini et al., 2019). For example, invariants, derived from the null space of the model's linear layers, lead to sets of inputs with identical

outputs. We refer to these sets as *equivalent sets*. Whereas nonsemantic invariants such as background or illumination are generally beneficial, invariants that carry semantic information may harm the classifier. However, although users can often introduce image augmentations to increase invariants of certain attributes, they cannot easily determine what the model has actually learned, only via rigorous testing.

This motivates approaches that interpret neural networks while focusing on their geometry. A natural starting point would be the geometry of the classification head, where the last decision is made. A related line of research applies singular value decomposition (SVD) to the latent space based on representative data in the latent feature space (Aubry & Russell, 2015; Härkönen et al., 2020; Haas et al., 2024); however, these methods are prone to the data covariances rather than network mechanism. Other methods operate directly in the weight-induced null space (Cook et al., 2020; Rezaei & Sabokrou, 2023; Li & Short, 2024). For example, the classifier head can be decomposed into two space components:(i) principal directions, associated with dominant singular values that influence the logits; (ii) null directions, the complementary space that keeps the inputs unchanged (Praggastis et al., 2022; Anthes et al., 2023). While they are able to identify the existence of invariant directions, they fail to explain semantically what they represent, and often rely on task-specific data to demonstrate these directions (Li & Short, 2024).

Recent advances in mechanistic interpretability (Moayeri et al., 2023; Kim et al., 2023; Huang et al., 2024; Dreyer et al., 2025) enable the translation of latent features from a given model into a multimodal vision language space, most notably CLIP (Radford et al., 2021). The use of CLIP to compute semantic correlations between text and images facilitates new sets of techniques that focus on producing human-readable concepts and counterfactual examples to aid interpretation. However, to the best of our knowledge, we are the first to map a classifier's invariant directions into a multi modal network for systematic analysis, providing textual descriptions and visual examples.

In this work, we propose the Semantic Interpretation of the Null-space Geometry (SING) method. Its purpose is to identify and explain representation of equivalent pairs in the latent feature space of a target classifier. We leverage SVD of the feature layer to capture hidden information in the null space. Interpretability is obtained by training linear translators to CLIP's space, allowing quantifiable statistical semantic analysis. Our method provides a general framework to measure human-readable explanations. This allows to probe, debug and compare data invariants from the image and class levels up to entire model assessments. It can be used to detect vulnerable classes and spurious information, such as background cues. We demonstrate the effectiveness of SING through cross-architecture measurements, per-class analysis, and individual image breakdown. In the last section of our experiments we present a promising direction for null space manipulation, creating features with hidden semantics that the model ignores. Our main contributions are:

- *A semantic tool for interpreting invariants*. SING links classifier geometry, specifically the null space and the invariants it induces, to meaningful human-readable explanations using equivalent pairs analysis.
- *Model comparison*. We introduce a protocol to compare different architectures by measuring the leakage of their semantic information into their null space. Our analysis found that DINO-ViT, among the examined networks, had the least class-relevant leakage into its null space while allowing broad permissible invariants, such as background or color.
- *Open vocabulary class analysis*. Our framework allows for systematic investigations of the sensitivity of classes to certain concepts. It can discover spurious correlations and assess their contribution. For example, our experiments show that for some spurious attributes in the DINO-ViT model the classifier head considers them as invariants.

## 2 RELATED WORK

### 2.1 EXPLAINABILITY THROUGH DECOMPOSITION

Decomposing latent spaces using SVD is a foundational approach for studying their invariances (Golub & Reinsch, 1970). Aubry & Russell (2015) used this technique to probe dominant modes of variation in CNN embeddings, for example illumination and viewpoint, under controlled synthetically rendered scenes. Härkönen et al. (2020) applied it to GAN latent spaces for interpretable

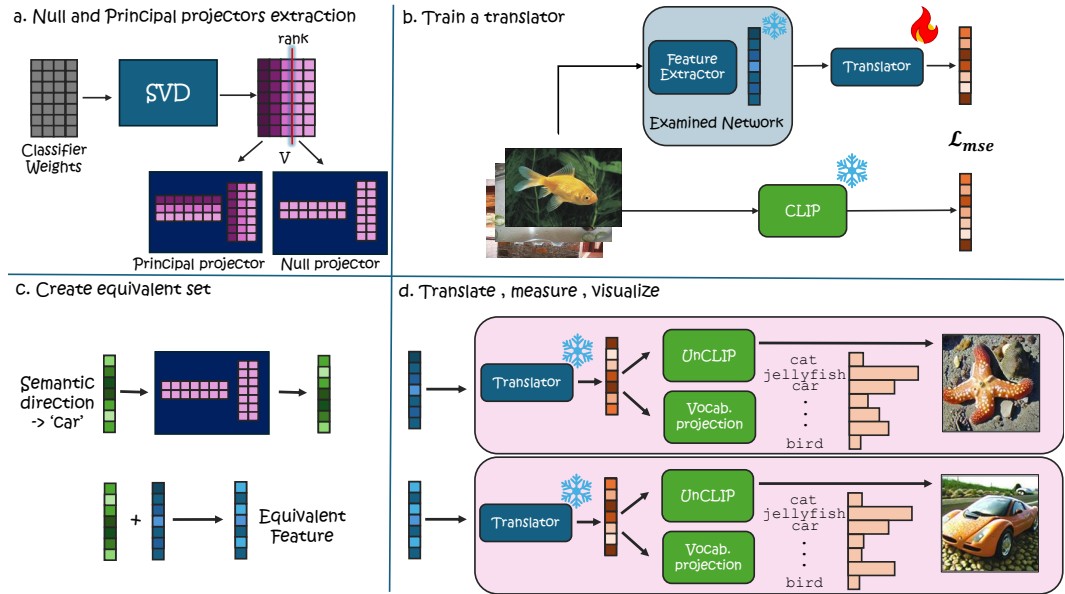

Figure 2: **Method Overview.** The approach consists of: (a) decomposing the final linear weights to obtain principal and null projectors; (b) training a translator that maps features from the network embedding space to the CLIP image space; (c) creating an equivalent pair to the feature we want to examine. (d) translate the set into CLIP image embedding space, and apply our metrics and visualizations.

controls, and more recently Haas et al. (2024) used it to present consistent editing directions in diffusion model latent spaces. However, feature-space decomposition is inherently data-dependent: its axes reflect the covariance of the measured dataset rather than the classifier's decision geometry; crucially, it may miss invariances residing in the classifier's null space itself.

A complementary study involves decomposing the model weights directly. This line of work includes early low-rank decompositions of convolutional weights for acceleration (Jaderberg et al., 2014), SVD analyzes of convolutional filters for interpretability (Praggastis et al., 2022), and decomposition of the final linear layer to identify the direction relevant to the task and the direction invariant to the task (Anthes et al., 2023). Ravfogel et al. (2020) iteratively projected representations onto the null space of a linear attribute classifier to remove protected information while preserving task predictions; Cook et al. (2020) performed null space analysis inside networks and use projections to derive an OOD detection score; Idnani et al. (2023) explained OOD failures via null-space occupancy, showing that features drifting into the readout's null space lead to misclassification; Rezaei & Sabokrou (2023) analyzed the last layer null space to quantify overfitting through changes in null space structure; and Li & Short (2024) exploited null space properties to perform image steganography, masking images that leave logits unchanged. Collectively, these methods treat the null space as an operational invariance set for control, detection, and manipulation. However, as far as we know, no current research managed to assign *semantic meaning* to null directions, as our approach does.

## 2.2 PROJECTING FEATURES TO A VISION-LANGUAGE SPACE

Contrastive Language–Image Pretraining (CLIP) (Radford et al., 2021) learns a rich joint embedding space for images and text, enabling a wide range of vision-language applications. However, its internal latent geometry remains poorly understood and exhibits a modality gap (Liang et al., 2022). This latent space has been analyzed both geometrically (Levi & Gilboa, 2025) and probabilistically (Betser et al., 2025). Several methods have leveraged CLIP representations for interpretability. For example, Text2Concept (Moayeri et al., 2023) learns a linear map from any vision model's latent space to CLIP's space so that text embeddings act as concept activation vectors, supporting zero shot concept queries without curated concept labels. CounTEX (Kim et al., 2023) similarly derives

concept directions from text in CLIP and introduces a projection and it's inverse, to project back and forth between the classifier and CLIP, generating counterfactual examples. Complementing these projection based approaches, LG CAV (Huang et al., 2024) uses CLIP as language guided supervision on unlabeled probe images to train concept vectors directly in the target model's feature space. In parallel, CLIP Dissect (Oikarinen & Weng, 2022) assigns open vocabulary concept labels to neurons in arbitrary vision networks by matching activation visualizations with CLIP embeddings, enabling label free network dissection without curated concepts. This vast research explicitly presents the benefits of probing a model. However, unlike our approach, these methods focus on manipulating the classifier's features and overlook the null space of these networks.

## 3 METHOD

Our method contains several components as can be seen in Figure 2. We begin by decomposing the target layer into principal and null subspaces and building projection operators that isolate each space. On the second component, we learn a linear mapping that translates the layer's features into the shared multi-modal space, specifically the image space. We then select a feature and perturb it along a specified semantic direction projected to a chosen subspace, creating the equivalent feature pair. After perturbing, we translate the feature using our translator to observe how its representation changed semantically with visualization and textual measurements. In this section we develop each component in detail, with particular attention to the null space and to the classifier head.

### 3.1 SETUP

In our work, we focus on the last fully connected layer $W \in \mathbb{R}^{c \times m}$, which maps the penultimate features $f \in \mathbb{R}^m$ to a logit vector in the dimension of the number of classes $c$. We decompose it with SVD and specifically extract the null space projection matrix $\Pi_\mathrm{n}$, which contains all the invariants of the layer. In the translation step we denote $T_\Theta(f)$ as the *Translator*, and we use CLIP as our multi-modal model space. We denote $z^{img}$ and $z^{text}$ as the image and text latent features in CLIP space. We define $\tilde{f}$ as the equivalent pair of $f$ after perturbation in the null space.

### 3.2 SVD ON THE CLASSIFIER HEAD

$W$ can be decomposed into its principal and null spaces via SVD:

$$W = U \, \Sigma \, V^\top, \qquad V = \begin{bmatrix} V_\mathrm{p} & V_\mathrm{n} \end{bmatrix}, \tag{1}$$

where $\Sigma \in \mathbb{R}^{c \times m}$ is a rectangle diagonal matrix containing the singular values in descending order, and $U \in \mathbb{R}^{c \times c}$ and $V \in \mathbb{R}^{m \times m}$ contain the left and right singular vectors, respectively. We take $\mathrm{rank}(W)$, and use it to break the right singular vectors $V$ into the two subspace components, *principal space*, denoted $V_\mathrm{p}$ (associated with non-zero singular values), and the remaining columns $V_\mathrm{n}$ that span the *null space*. Any perturbation $\nu \in \mathrm{span}(V_\mathrm{n})$ leaves the logits unchanged:

$$W(f + \nu) = Wf + W\nu = Wf, \tag{2}$$

since $W\nu = 0$ for all $\nu$ in the null space. Consequently, our projector matrices are:

$$\Pi_\mathrm{p} = V_\mathrm{p} V_\mathrm{p}^\top, \qquad \Pi_\mathrm{n} = V_\mathrm{n} V_\mathrm{n}^\top. \tag{3}$$

### 3.3 TRAINING A TRANSLATOR

Following Moayeri et al. (2023) and justified by Lähner & Moeller (2024), we define a linear mapping operator $T : \mathbb{R}^m \to \mathbb{R}^n$. Recall that $f \in \mathbb{R}^m$ is the classifier feature and $z^{img} \in \mathbb{R}^n$ the corresponding image feature in CLIP. We fit $T_\Theta$ for a certain pretrained model by minimizing a loss combining mean squared error, and weight decay:

$$\mathcal{L} = \|T_\Theta(f) - z^{img}\|_2^2 + \lambda \, \|\Theta\|_2^2, \tag{4}$$

where $\Theta$ is the parameters of the translator and $\lambda$ is a balancing coefficient. Detailed explanations on the training procedure can be found in Appendix B. Note that since the translator is linear, it admits

$T_\Theta(f + v) = T_\Theta(f) + T_\Theta(v)$ for any $f, v$, hence naturally fits additive feature decompositions, as our framework suggests.

## 3.4 METRICS

**Attribute score.**  An angle between two nonzero vectors $x, y$ of the same dimension is defined by:

$$\angle(x, y) := \arccos\left(\frac{x \cdot y}{\|x\|\|y\|}\right). \tag{5}$$

CLIP Score, as described in Hessel et al. (2021), is the cosine similarity of the angle between a CLIP feature in image space $z^{img}$, and a feature in the text space, $z^{text}$. We write this angle as follows:

$$\angle(z^{img}, z^{text}) \tag{6}$$

Recall that $f$ and $\tilde{f}$ are the original and its equivalent pair. We define *Attribute Score* (AS) for text target $z^{text}$ as the difference between two angles:

$$\text{AS}(f, \tilde{f} | z^{text}, T_\Theta) := \angle(T_\Theta(f), z^{text}) - \angle(T_\Theta(\tilde{f}), z^{text}). \tag{7}$$

A positive AS indicates that the equivalent image is semantically closer to the text and vice versa. In our framework, the text prompts are chosen as "`an image of a <class>`" to analyze how null removal affects classification. However, this metric is general and can be applied with any prompt selection.

**Image score.**  While AS quantifies how the image deviates from its current semantics, the image may be altered in appearance without affecting AS. Such differences in overall appearance can be measured directly by the angular distance related to the original and its equivalent pair. we define it as *Image Score* (IS):

$$\text{IS}(f, \tilde{f} | T_\Theta) := \angle(T_\Theta(f), T_\Theta(\tilde{f})). \tag{8}$$

Intuitively, AS captures the effects of null spaces on the alignment of text-image, whereas IS reflects general semantic changes in the image. When the text is in the correct image class we would like low AS, and hence null-space changes should not affect class distinction. However, a good classifier should allow high IS, and hence large semantic changes that do not affect class distinction, such as background change and other allowed semantic invariants. Details on image synthesis for visualization are provided in Appendix D.

## 3.5 APPLICATIONS

Our main focus is on removing the null component from an image feature $f$. This way, the equivalent pair is

$$\tilde{f} = f - \Pi_\text{n} f. \tag{9}$$

Both $f$ and $\tilde{f}$ produce the same logit vector under the examined network, yet the semantic content can be changed as a result of the null-removal process. In the following, we describe how to quantify semantic information leakage at different levels: model, attribute, and image, using the proposed metrics (AS and IS).

**Model-level comparison.**  A desirable property of well-performing classifiers is to maintain a rich invariant space, while ensuring that this richness does not compromise class preservation. For instance, there exists a wide variety of dogs differing in breed, pose, size, color, background and more, all of which should be classified consistently with high confidence. Hence, the invariant space should support such diversity. However, if perturbations along invariant directions lead to changes in classification confidence or even alter the predicted class, this indicates that class-specific information has leaked into the invariant space - a highly undesirable property that also exposes the model to adversarial vulnerabilities. To evaluate this, we collect a representative set of images (16 ImageNet classes, serving as a proof of concept), compute the AS and IS metrics (with respect to the real class prompt; "`an image of a <ground-truth class>`") on all null-removed pairs, and perform a statistical analysis across models. An effective model should exhibit a broad range of IS values, reflecting rich invariance, while maintaining a narrow distribution of AS values, ensuring semantic consistency.

**Class and Attribute analysis.** The same methodology can be applied to analyze inter-class behavior by selecting representative sets from different classes. We conducted two complementary variants. First, we collected images from each class independently and computed the absolute Attribute Score (AS) after null-removal, relative to the true label prompt. Higher AS values indicate that the classifier contains more semantic information within the invariant space for that class. This provides a practical diagnostic tool for practitioners when choosing networks suited to specific classes or domains. Second, we expanded the vocabulary to an open set of concepts. We quantified the distance (angles) between the original and the null-removed features, over a broad set of phrases, revealing how semantic correlations emerge between the null space and diverse concepts.

**Single image analysis.** Following the same logic, leakage can also be examined at the image level. This provides a fine-grained diagnostic tool for identifying and debugging failure cases.

**Null perturbations.** While null removal is useful for fair comparisons across classes, attributes, or images, feature manipulation need not be restricted to a single invariant direction. We propose a more principled selection of perturbation directions. We formalize perturbations that target a specific concept while remaining confined to the model's invariant (null) subspace. Let $f \in \mathbb{R}^d$ be an image feature, $T_\Theta : \mathbb{R}^d \to \mathbb{R}^n$ the translator into the CLIP image-embedding space, and $z_{\text{text}} \in \mathbb{R}^n$ the CLIP text embedding of a prompt (e.g., "`an image of a jellyfish`"). Define the cosine-similarity score

$$s(f; z_{\text{text}}) := \frac{\langle z, z_{\text{text}} \rangle}{\|z\| \, \|z_{\text{text}}\|}, \qquad z := T_\Theta(f). \tag{10}$$

The *semantic direction* toward the prompt is the gradient through the translator,

$$g_{\text{text}}(f) := \nabla_f \, s(f; z_{\text{text}}). \tag{11}$$

Let $\Pi_{\text{n}}$ denote the orthogonal projector onto the null space (equation 3). Projecting this direction onto the null space isolates the component that lives in the invariant subspace:

$$d_{\text{null}}(f) := P_\mathcal{N} \, g_{\text{text}}(f), \qquad \hat{d}_{\text{null}}(f) := \frac{d_{\text{null}}(f)}{\|d_{\text{null}}(f)\|}. \tag{12}$$

One can control the extent of semantic change via a scalar step size $\varepsilon$ applied to the normalized null direction $\hat{d}_{\text{null}}$:

$$f_\varepsilon = f + \varepsilon \, \hat{d}_{\text{null}}(f). \tag{13}$$

By choosing the prompt to correspond to another class or attribute, this construction probes a class's sensitivity *within* the invariant subspace to concepts associated with other classes, thereby revealing "confusing" inter-class relationships.

## 4 EXPERIMENTS

### 4.1 DATASET AND MODELS

We base our analysis on five models pretrained on ImageNet-1k (Deng et al., 2009) spanning diverse architectures and training paradigms: DINO-ViT (Caron et al., 2021), ResNet50 (He et al., 2016), ResNeXt101 with weakly supervised pretraining (Mahajan et al., 2018), EfficientNetB4 trained with Noisy Student (Xie et al., 2020), and BiTResNetv2 (Kolesnikov et al., 2020). For statistical analyses, we collect 10k feature vectors per model from a restricted subset of 16 classes. For each model, we then train a dedicated translator; to keep the study focused on a proof of concept, translator training is limited to the same representative subset of 16 classes. Additional details appear in Appendix B.

### 4.2 MODEL COMPARISON

We compare models globally across all tested classes, measuring AS and IS after null removal. Figure 3 displays the joint distributions of AS and IS across five models. DINO-ViT attains the best IS/AS trade-off, consistent with its foundation-scale pretraining on a large, diverse corpus beyond ImageNet prior to fine-tuning. This trade-off is evident both in the IS/AS ratio bar plot (panel b)

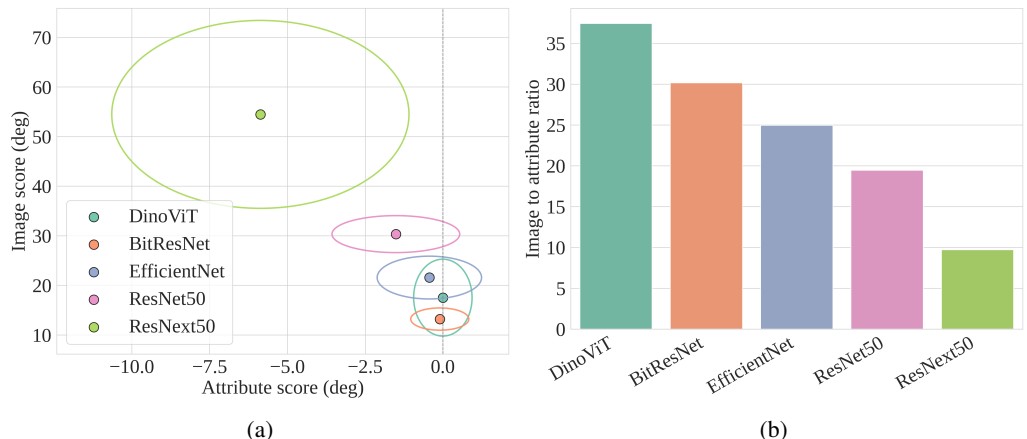

(a)                         (b)

Figure 3: **Model-level comparison.** (a) Attribute Score (AS) quantifies *class-dependent* semantic leakage into the null space; Image Score (IS) quantifies tolerance to *class-independent* (non–class-dependent) semantic variation within the invariant subspace. Desirably, AS is low and IS is high (relative to AS). In our results, DINO-ViT performs best in this regard, having the most vertically elongated confidence ellipse. (b) We summarize the trade-off with the IS/AS ratio (higher is better), DINO-ViT has the highest ratio and ResNeXt101 the lowest.

and in the orientation of the confidence ellipses in panel (a): only DINO-ViT shows vertical elongation, i.e., $\text{Var}(\text{IS}) > \text{Var}(\text{AS})$, whereas all other models exhibit the opposite pattern. By contrast, ResNeXt101 shows high AS with substantial variance, which we interpret as class-dependent semantic leakage into its null space. As discussed in Section 3, such leakage suggests a geometric vulnerability in the model.

### 4.3 CLASS ANALYSIS

We present per class statistics of AS for two of our models, ResNet50 and DINO-ViT , and report them class by class; see Figure 4. For each class, AS is measured after null removal. A complete analysis of the other models can be found in Appendix F DINO-ViT exhibits stable behavior with very small AS magnitudes (typically $|\text{AS}| < 1$), consistent with minimal class-dependent leakage into the null space. By contrast, ResNet50 shows larger and more variable AS across classes. This contrast suggests that DINO-ViT tends to retain class-relevant semantics within its invariant subspace, whereas ResNet50 appears to possibly rely also on spurious cues, leaving some class-relevant information in the null space. Finally, we observe no significant correlation between the per-class AS rank orderings of the two models, indicating that the effect is model-dependent rather than driven by dataset class structure.

In fig. 5, We extend the class analysis to an open vocabulary of concepts. Focusing on DINO-ViT, we examine two classes, "Arabian Camel" and "Jellyfish". We measure two quantities: 1) The *angle* between the translated feature and the CLIP concept embedding; 2) the Attribute Score (AS), quantifies how much content related to a concept resides in the null space; A small AS for loosely related concept can indicate a spurious correlation. Both classes are analyzed through a set contains of 30 concepts, the extreme weakest and strongest are presented. "Arabian Camel" features exhibit little to no AS (short green lines), while Desert attains the smallest CLIP angle among the tested concepts. By contrast, "Jellyfish" features have substantially larger AS, indicating that concepts are tightly coupled to invariances related to this class in the classifier head. The results on the full set of open-vocabulary concepts are in Appendix G, and intuition for the scale of AS values is provided in Appendix C.

### 4.4 GRADIENT DIRECTION ANALYSIS

In the previous experiments, we restricted our analysis to equivalent pairs obtained by removing the null component. However, our method supports any null-space direction, including text-conditioned

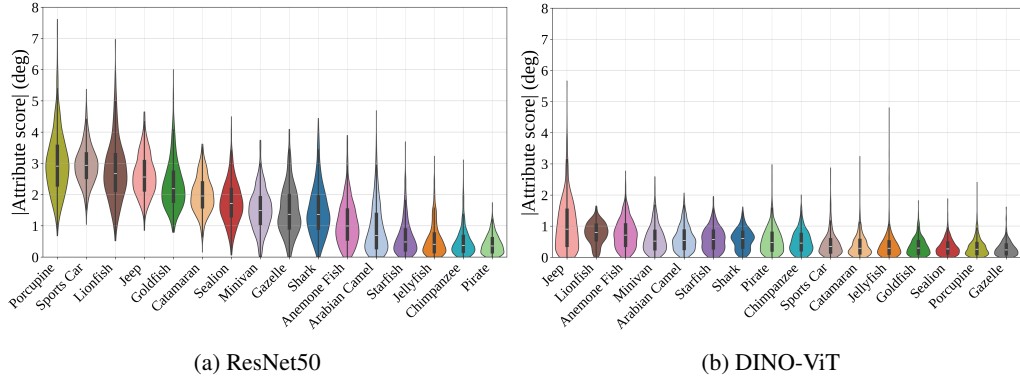

(a) ResNet50          (b) DINO-ViT

Figure 4: **Class Comparison.** DINO-ViT consistently preserves low semantic leakage across classes, whereas ResNet50 exhibits a pronounced imbalance, with certain classes, such as Porcupine and Sports-Car, leaking substantially more semantic information into the null space.

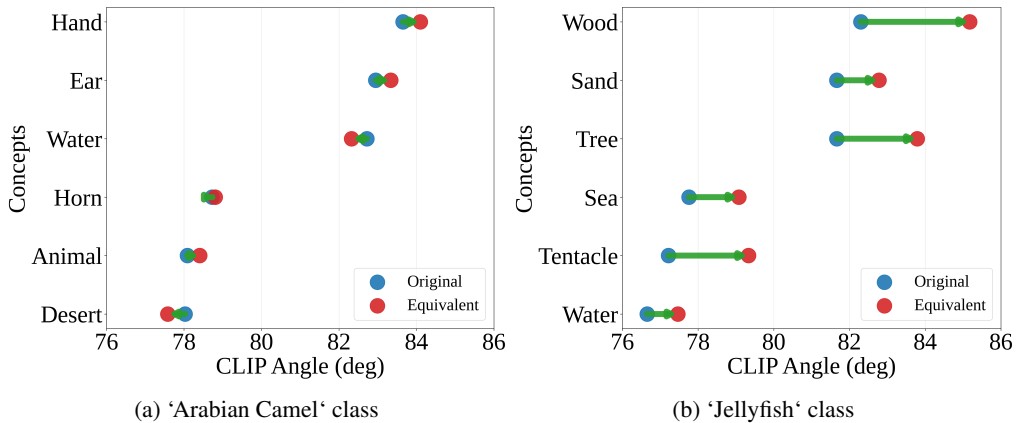

(a) 'Arabian Camel' class        (b) 'Jellyfish' class

Figure 5: **Open-vocabulary concept analysis.** For DINO-ViT, we sample $\sim 1300$ images per class and compute the CLIP angle (degrees; lower is more similar) to a set of concepts for (a) "Arabian Camel" class and (b) "Jellyfish" class. Blue dots denote original features; red dots denote null-removed (equivalent) features. Green arrows connect each pair and represent the Attribute Score after null removal. Longer arrows indicate larger |AS| (greater class-dependent semantic leakage); shorter arrows indicate minimal leakage.

perturbations. In Figure 6, we illustrate concept-directed perturbations confined to the null space of the ResNet50 classifier head. For each original image (left), we follow the CLIP similarity gradient toward a target prompt, project it onto the null space, and take a step in this direction to obtain an equivalent feature. By construction, the perturbed feature leaves the head logits unchanged. The synthesized renderings, generated with UnCLIP (Ramesh et al., 2022) for visualization, reveal pronounced semantic shifts toward *Arabian Camel*, *Starfish*, *Pirate*, *Jellyfish*, and *Jeep*. This demonstrates the diagnostic value of null-space steering and highlights a security risk: semantics can be manipulated at a single layer while the classifier's decision remains unaffected.

Table 1 summarizes null-space steps (calibrated to IS = 40°) from *Sports Car* toward the prompt "an image of a jellyfish". In this setting, DINO-ViT exhibits low AS, indicating resilience to directed null manipulation. By contrast, EfficientNet and ResNet50 show large AS, suggesting that their null components are easier to steer and that directed invariant perturbations can alter semantics while leaving the logits unchanged.

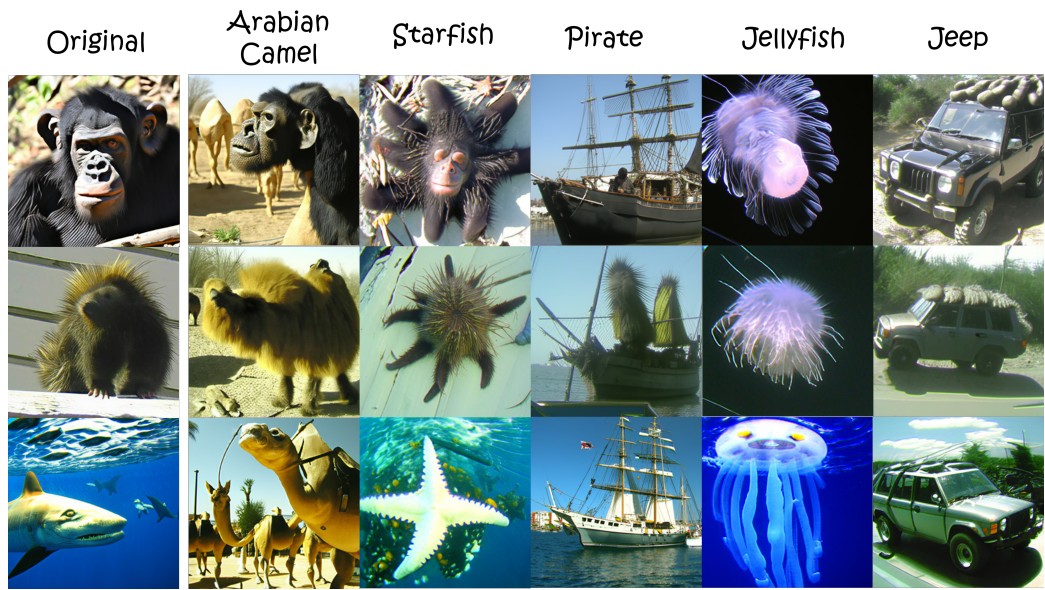

Figure 6: **Null-space semantic steering (ResNet50).** From each original image (left), we add a small perturbation aligned with the indicated prompt (column headers) but constrained to the classifier head's null space (projected-gradient direction). Although only the invariant component is modified, the feature's semantics shift toward the target concepts, illustrating how null-space directions can alter meaning without changing the discriminative subspace.

Table 1: **Text-gradient null perturbations.** For a fair comparison, each model is perturbed by a fixed null-space step calibrated to IS = $40°$. We report |AS| toward the target prompt (mean $\pm$ standard deviation; lower is better). DINO-ViT attains the lowest value (marked in bold), indicating the greatest resistance to directed null-space manipulation, whereas ResNeXt101 remains comparatively susceptible.

|  | ResNet50 | EfficientNet | BiTresnet | DINO-ViT | ResNeXt101 |
|---|---|---|---|---|---|
| \|AS\| towards target | 12.04±0.25 | 12.38±0.52 | 9.19±0.31 | **5.0±0.59** | 11.15±0.53 |

## 5 DISCUSSION AND CONCLUSION

We introduced SING, a novel approach for analyzing invariances in classification networks. Our method systematically generates equivalent images whose logits are, by construction, identical to those of the original image. We demonstrated a wide range of possible analyses: at the model level, SING facilitates fair sensitivity comparisons across architectures; at the class level, it highlights classes that are less robust to semantic shifts; and at the image level, it aids in debugging failure cases. SING transforms the null space into measurable and human-readable evidence by constructing equivalent pairs, projecting features into a joint vision-language space, and perturbing only the invariant component. In doing so, it reveals how semantics can drift while logits remain fixed, providing a compact diagnostic that complements accuracy at the levels of models, classes, and individual images. Looking ahead, two research directions may help control the null space more directly: (i) Directed augmentation during fine-tuning, encouraging small AS for essential concepts; (ii) Linear-algebraic control, using projector regularization, rank adjustment, or constrained updates to move useful semantics from the null space to the principal space while preserving logits. SING exposes invariant geometry in a simple, interpretable form, clarifying how semantics can shift while logits remain fixed.

## REPRODUCIBILITY

We instantiate each model with official pre trained weights and the evaluation transforms provided by the corresponding weight objects. ResNet50 uses ImageNet1K V1 weights from the `torchvision` package. DINO-ViT uses ViT-L/16 with ImageNet1K SWAG E2E V1 with their official `torchvision` weights. EfficientNet B4 uses the `timm` with B4 Noisy Student JFT weights, ResNeXt101 32x8d uses the `timm` FB WSL IG1B weights fine tuned on ImageNet1K, and BiT ResNetV2 50x1 uses the `timm` BiT weights pre trained on ImageNet21K and fine tuned on ImageNet1K. All models and transforms are from PyTorch and timm (Paszke et al., 2019; Wightman, 2019). In all cases $f$ denotes the penultimate feature just before the final linear head that produces logits, and $W$ denotes that head weight matrix; these define the SVD and the principal and null projectors. Details of the DINO-ViT wrapper appear in Appendix E.

## ETHICS STATEMENT

We affirm that all authors have read and will adhere to the ICLR Code of Ethics. Our study uses publicly available ImageNet data and pretrained models; no new data collection was conducted, no human subjects research was performed, and no personally identifiable information was processed. Because CLIP and ImageNet can encode social and geographic biases, we report concept-based analyses with care, avoid protected attributes in our concept lists, and interpret results in light of known dataset biases. All data and model use complies with source licenses for PyTorch and timm; we do not redistribute restricted assets. Compute was kept modest by relying on existing checkpoints and small translators. There are no conflicts of interest or external sponsorship beyond what is acknowledged in the paper.

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

## A   USAGE OF LLM

We used large language models to improve the writing and to assist the literature search. Research LLM used to surface candidate papers, but we independently verified their relevance, and retained only sources we judged necessary. Furthermore, we did not limit the paper search to LLM and used other tools as well. For writing, we used LLM based rephrasing to raise the level of English while ensuring that the technical meaning remained unchanged.

## B  TRANSLATOR TRAINING PROCEDURE.

We trained the translators using PyTorch Paszke et al. (2019). We combined 2 losses testing MSE and cosine similarity. MSE alone proved enough to satisfy cosine similarity as well, while the opposite did not. To complete the ablation we added a joint loss with similar weights for the MSE and the cosine similarity as can be seen in Figure 7. This phenomenon is expected since cosine similarity defines the angle between two vectors and do not take in account the magnitude of the vectors. The translator model consists of 2 fully connected layers with no bias to retain stability. Using AdamW optimizer with strong weight decay coefficient ($\lambda = 0.1$) and learning rate of $1e-4$ proved to make the training faster and more stable for all translators (Loshchilov & Hutter, 2017). The final results of each translator can be viewed in Figure 8.

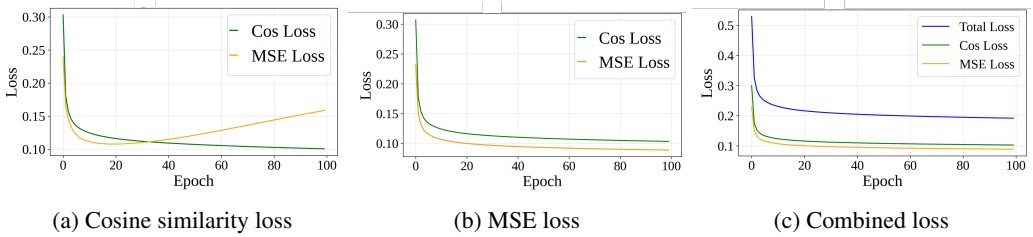

(a) Cosine similarity loss        (b) MSE loss        (c) Combined loss

Figure 7: Loss plots. It can be seen that while MSE manages to improve both scores, the cosine similarity by itself did not improve the MSE score as expected.

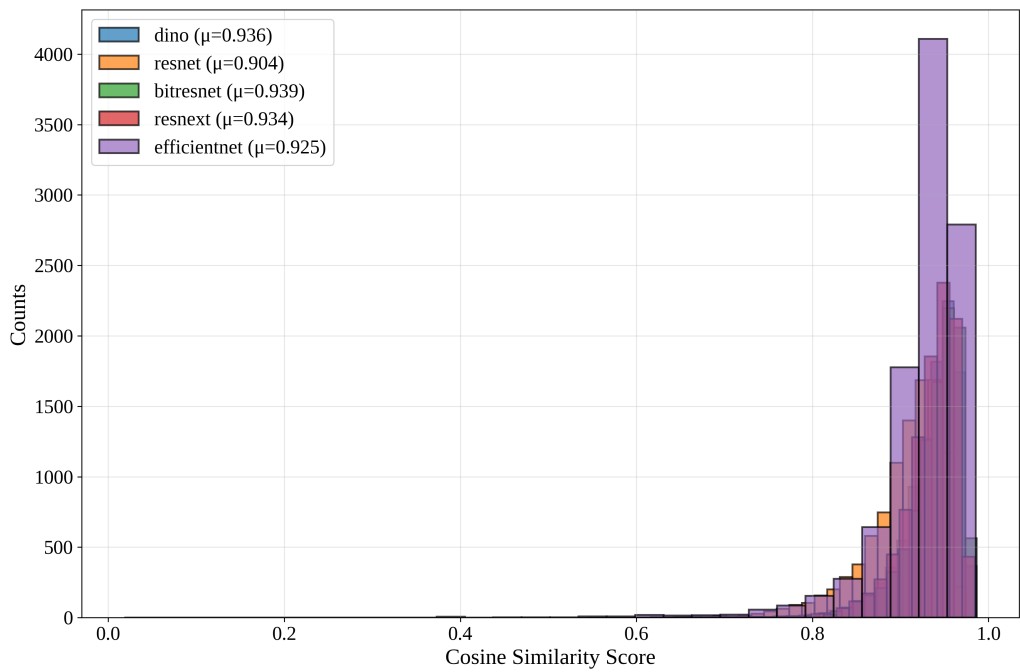

Figure 8: Evaluation of cosine similarity for 10K features from different classes on all translators. could not split the val outside so it's a mishmash of train and val together. but the concept of the graph stands. i have also loss figures if needed

## C  INTERPRETAION OF ANGLES TO VISULIZATION

Figure 9 provide a mapping of the different scores to grasp intuition of how much angle is considered semantic or differentiable. One can note that a semantic change occurs around $3°$ in attribute score and around $10°$ in image score.

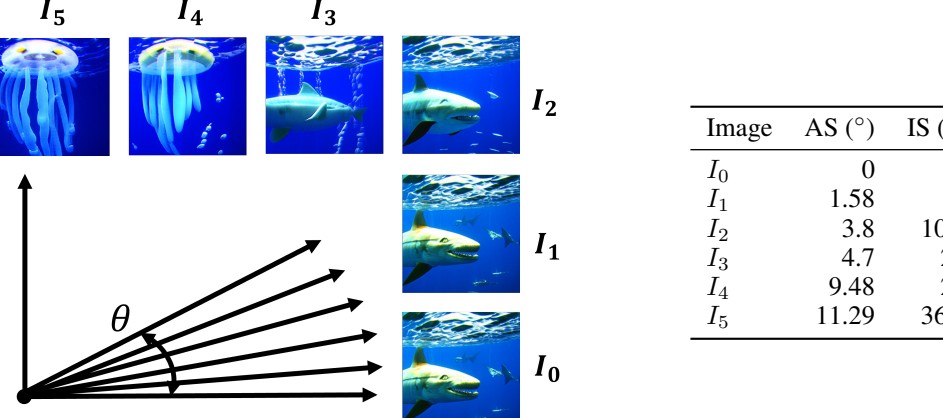

| Image | AS (°) | IS (°) |
|-------|--------|--------|
| $I_0$ | 0 | 0 |
| $I_1$ | 1.58 | 4 |
| $I_2$ | 3.8 | 10.8 |
| $I_3$ | 4.7 | 23 |
| $I_4$ | 9.48 | 29 |
| $I_5$ | 11.29 | 36.2 |

Figure 9: Different attribute score and image score levels, enhancing the intuition between the size of the angle and its matching visualization.

## D  VISUALIZATION WITH UNCLIP

UnCLIP is a two-stage generator: a *prior* maps text to a CLIP image embedding, then a diffusion-based *decoder* with super-resolution modules synthesizes the corresponding image (Ramesh et al., 2022). To our knowledge, we are the first to use classifier features, mapped in the CLIP image embedding space via a trained translator $T$, as a basis for target subspace traversal and visualization in model interpretability. By traversing $f$ along a chosen semantic direction $d$ (e.g. a null-space direction) and using shared initial noise, we obtain reproducible deterministic visualizations of the induced semantic changes. Empirically, we find that preserving the $\ell_2$-norm of the CLIP image embedding yields higher-quality generations for analysis and debugging. Given a feature and its equivalent feature set translated to CLIP: $T_\Theta(f), T_\Theta(\tilde{f})$, we rescale the equivalent feature as follows:

$$\hat{T}_\Theta(\tilde{f}) = T_\Theta(\tilde{f}) \frac{\|T_\Theta(f)\|_2}{\|T_\Theta(\tilde{f})\|_2}. \tag{14}$$

Since CLIP normalizes the embeddings of images and text to unit length and compares them through cosine similarity, semantic information is primarily encoded in the angular component off the unit hypersphere (Radford et al., 2021). Restoring the original norm retains the radial component without altering angular relationships, preventing distortions in the visualizations due to radial drift.

To ensure that observed visual differences are solely attributable to changes in the classifier feature $f$, we eliminate the stochasticity of the diffusion process sampling by reusing identical Gaussian noise in both the decoder and superresolution stages. Specifically, a single noise tensor is generated using `randn_tensor`, scaled by the scheduler's `init_noise_sigma`, and replicated across the batch for each stage. This procedure yields deterministic output for a fixed CLIP image embedding.

Our implementation employs the **Karlo-v1.0.alpha** UnCLIP model (Donghoon et al., 2022), based on the original OpenAI framework (Ramesh et al., 2022). It includes standard components: frozen CLIP text and image encoders, a projection layer, a `UNet2DConditionModel` decoder, two `UNet2DModel` super-resolution networks, and `UnCLIPScheduler` instances for both stages.

# E   DINO-ViT FEATURE WRAPPER

We expose the token sequence before the head, take the class token as the penultimate feature $f$, and use the head weights as $W$.

```
class SelectClassToken(nn.Module):
    def __init__(self, f):
        super().__init__()
        self.f, self.B = f, 1
    def forward(self, x):
        return x.reshape(self.B, -1, self.f)[:, 0, :]
    def set_B(self, B=1):
        self.B = B

class DinoHookable(nn.Module):
    def __init__(self, base: nn.Module, extractor, feature_dim=1024):
        super().__init__()
        self.extractor = extractor
        self.fc = base.heads.head
        self.penultimate = SelectClassToken(f=feature_dim)
    def forward(self, x: torch.Tensor) -> torch.Tensor:
        self.penultimate.set_B(x.size(0))
        x = self.extractor.extract(x, "encoder.ln")
        x = self.penultimate(x)            # penultimate feature f (class token)
        return self.fc(x)            # logits, head weight matrix is W
```

# F   COMPLETE VIOLIN ANALYSIS

We provide the violin analysis figures for all models that participated in our experiments in Figures 10 to 12

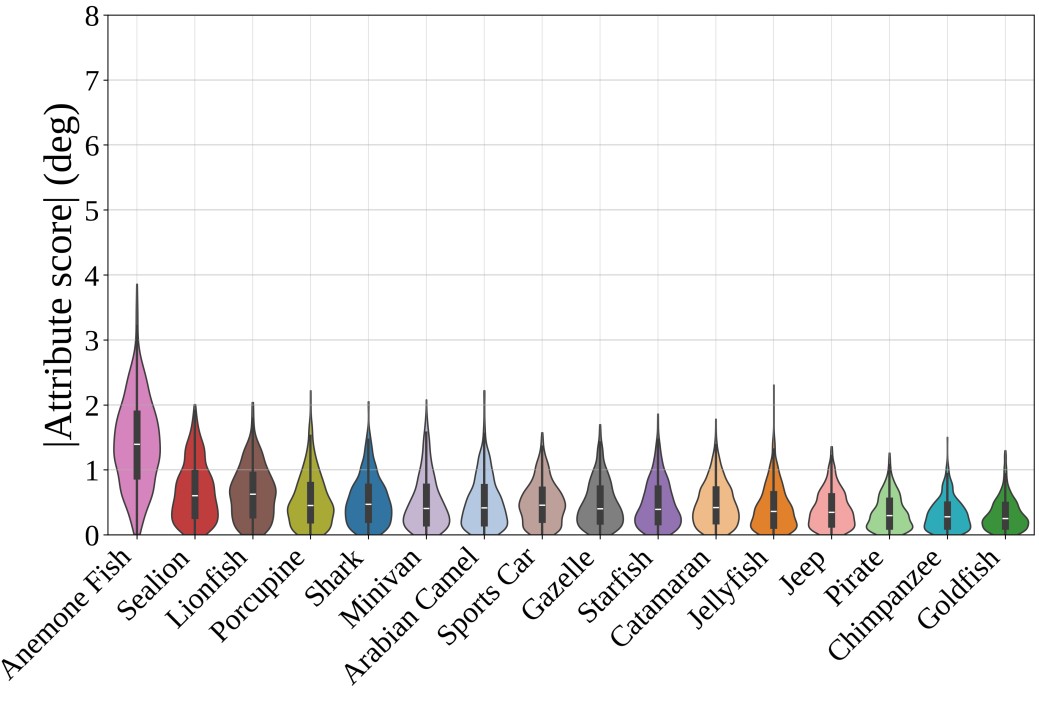

Figure 10: BiTresnet class analysis

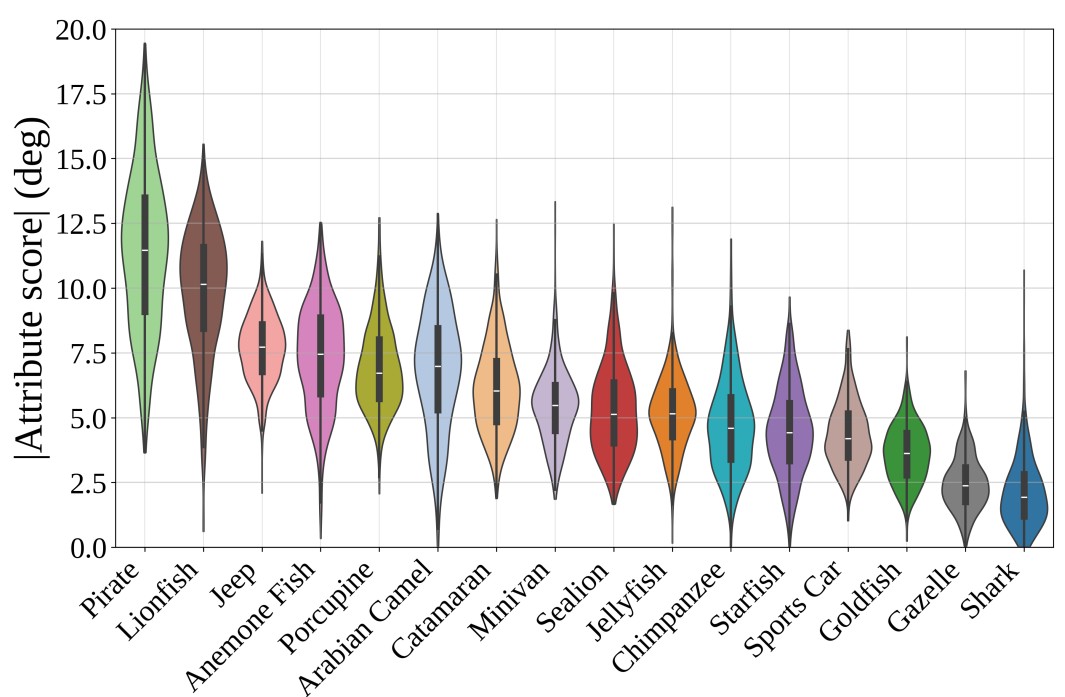

Figure 11: ResNeXt101 class analysis

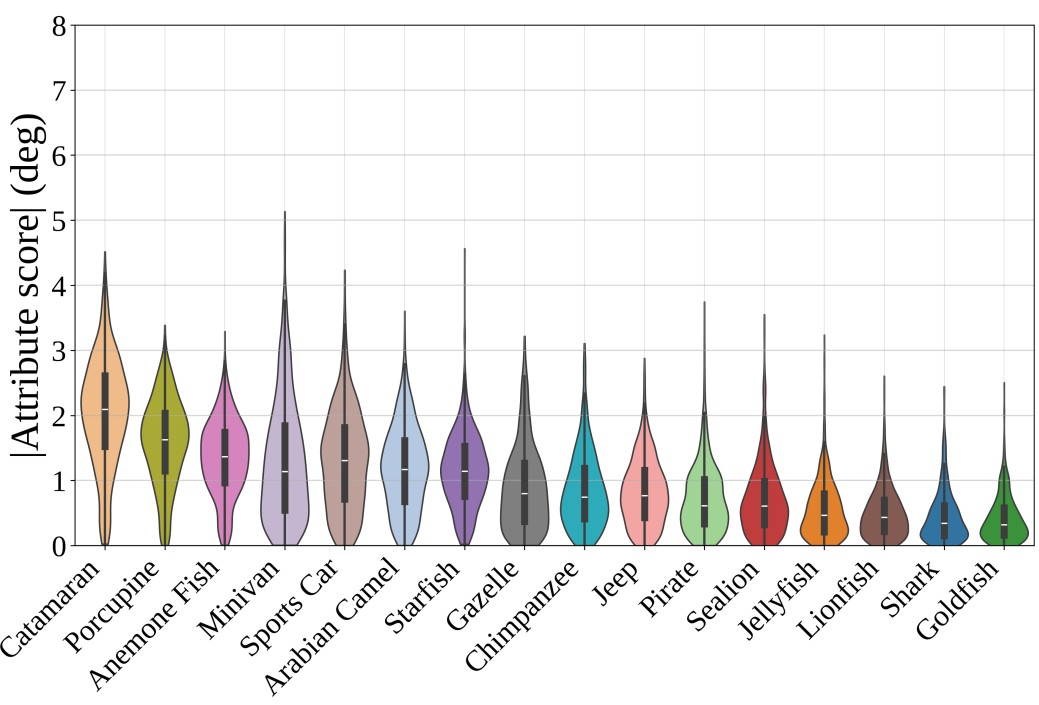

Figure 12: EfficientNet class analysis

## G COMPLETE CLASS ANALYSIS

Figures 13 and 14 provide a larger list of open vocabulary concepts that were used in the class analysis of the "Jellyfish" and "Arabian Camel" in DINO-ViT.

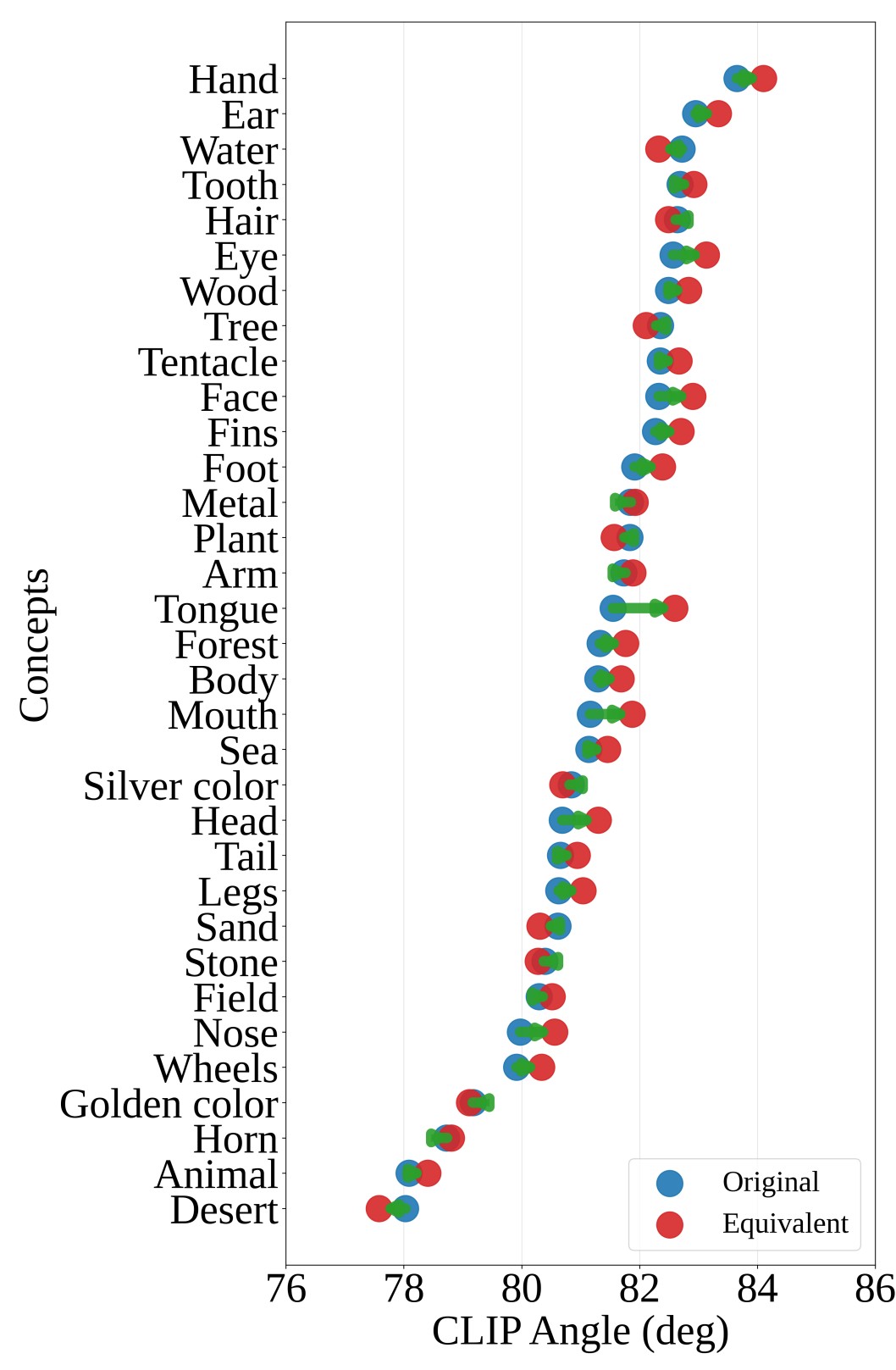

Figure 13: Open vocabulary analysis on "Arabian Camel" class in DINO-ViT

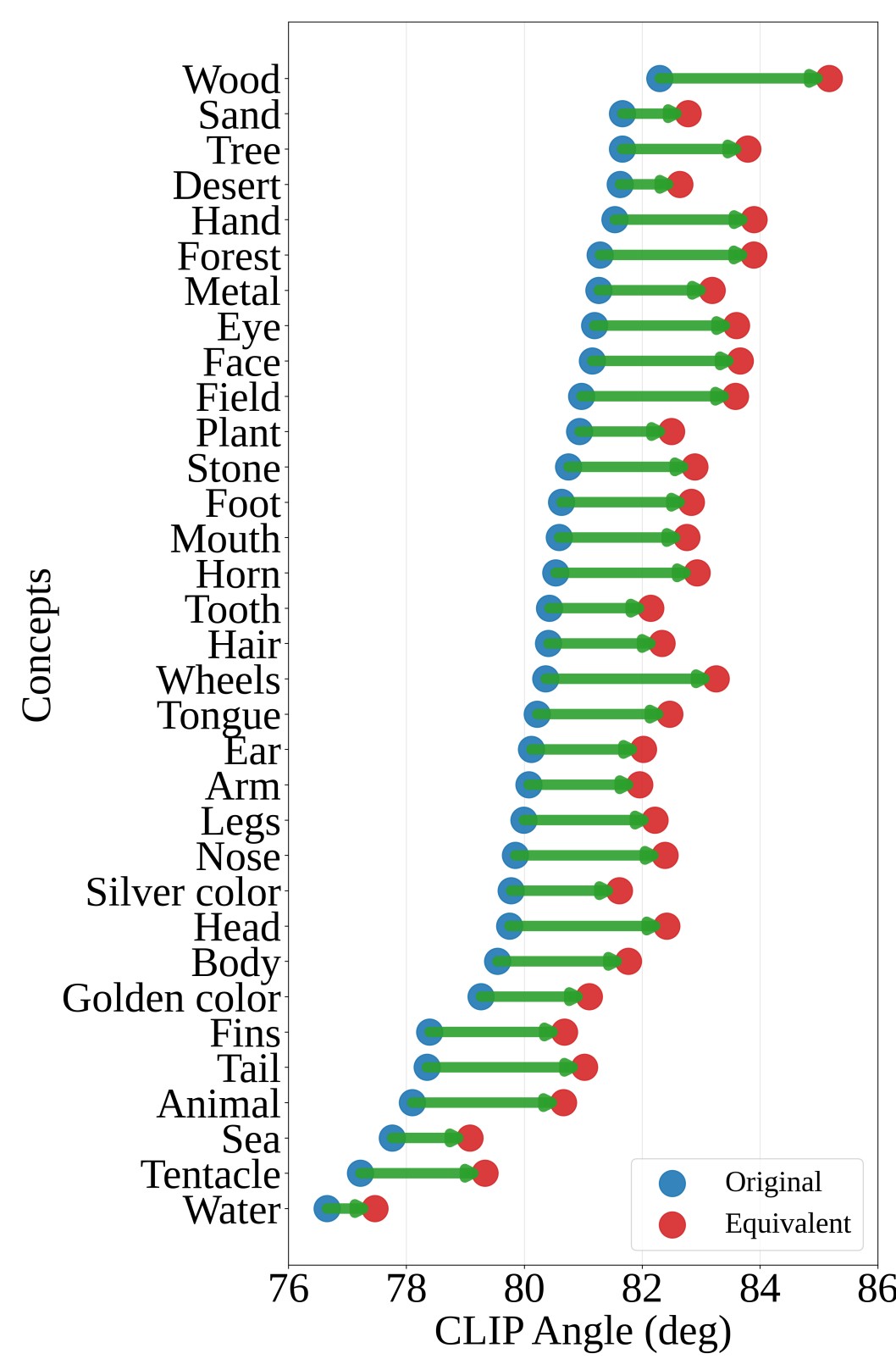

Figure 14: Open vocabulary analysis on "Jellyfish" class in DINO-ViT

