# OpenReview forum: "Make it SING: Analyzing Semantic Invariants in Classifiers"
_ICLR.cc/2026/Conference — ICLR 2026 Conference Withdrawn Submission_

### Official Review · Reviewer_JBLY · 2025-10-28

**Soundness:** 3
**Presentation:** 3
**Contribution:** 2
**Rating:** 6
**Confidence:** 3

**Summary:**

This paper introduces SING (Semantic Interpretation of the Null-space Geometry), a method to interpret the invariant directions of vision classifiers—the inputs that yield identical outputs due to the network’s null-space structure. By mapping these invariants into a multi-modal vision–language space such as CLIP, SING generates human-readable textual and visual interpretations of what information the model ignores. The framework enables both per-image and class-level analyses, revealing semantic leakage and providing new insights into model robustness and spurious correlations.

**Strengths:**

- The analysis is sound in theory, and the presentation is very clear and interesting.
- It is very nice to see the experiment on perturbed images.
- The results are solid and the experiments are interestingly designed.

**Weaknesses:**

- I wonder how is this method different from other concept based methods such as TCAV [1]? And also any other concept bottleneck methodologies [2]. I believe a similar result or analysis could be done by applying those methods.

- The work should have included more base ViTs such as DeiT [3] or CAIT [4] that is not trained within the vision language space. The author should also consider swin transformer [5].

- It would be interesting to extend the analysis into the class of Diffusion models.

[1] Interpretability Beyond Feature Attribution: Quantitative Testing with Concept Activation Vectors (TCAV)

[2] Concept Bottleneck Models

[3] Training data-efficient image transformers & distillation through attention

[4] Going deeper with Image Transformers

[5] Swin Transformer: Hierarchical Vision Transformer using Shifted Windows

**Questions:**

Please refer to the Weakness section

---

### Official Review · Reviewer_rT5f · 2025-10-28

**Soundness:** 2
**Presentation:** 3
**Contribution:** 3
**Rating:** 4
**Confidence:** 3

**Summary:**

This paper introduces a new method for interpreting vision classifiers by analysing the semantic content of the null space, which are the the directions in the feature space that the final linear layer ignores. It creates equivalent feature pairs that produce identical outputs and then translates them into CLIP's vision-language space to measure how the image's meaning has changed. The work presents interesting directions for model analysis and diagnostic work. The findings also produce some interesting directions for future research to understand the semantic leakage concept of the null space. However, the analysis is limited in places, primarily providing a heavy reliance on CLIP as a trusted oracle, and limited subset of semantic classes.

**Strengths:**

- The rationale is well defined and supported by previous works and some analysis of existing methods.
- SING provides an interesting and valuable direction of research into model evaluation, assessment and interpretation that while simple, could be a nice contribution to the field.
- The approach to study the null space for analysis the semantic meaning is a particularly novel and interesting direction that in this case seemingly provides unique and meaningful analysis of learnt model embedding spaces.
- The findings present an interesting direction for future research to understand the reduces semantic leakage of the ViT based model. Thus presenting a significant contribution to improve modelling of semantic concepts.
- Limitations, reproducibility and ethics are presented and discussed.

**Weaknesses:**

**Major:**

- From my understanding the proposed method relies on the correctness of the CLIP encoder, meaning that it is assumed that CLIP is effective at appropriately structuring the embedding in a semantically meaningful way. If CLIP is insufficient the method itself may before poorly. Some analysis of this issue would be a good study for robustness and generalisation to other vision-language models at different performance.
- The rationale and analysis behind the metrics could be elaborated, these essentially measure the closeness in embedding space, but do not take into account semantic hierarchies or composition of concepts. More advanced graph-based or representation similarity rather than embedding angle.
- Some formal or empirical analysis of the Null perturbations would be expected to guarantee that such perturbation is operating under the expected behaviour.
- The analysis for model comparison is performed on only 16 out of 1000 ImageNet classes.
- The authors approach provides a “local” analysis of the model's decision looking at only the final linear layer, not a comprehensive understanding of how the entire network processes semantic information. Thus, how the full extent of semantics through the entire network features is ignored.

**Minor:**
- Arguably, Figure 2 could be more descriptive of the method, or the caption itself be improved. The flow of the method is not fully understandable, hence limiting the relevance of the figure.
- See questions for other minor weaknesses.

**Questions:**

1. Does the method rely on the validity and correctness of the CLIP? Simply put if CLIP produces invariants your translator seemingly will do the same?
2. Could it be that the ViT based network (DINO) demonstrates reduced semantic leak as the CLIP encoder also uses a ViT and hence the shared architectural base has an influence?
3. Is the linear map T expressive enough to capture the translation complexities?

---

### Official Review · Reviewer_Eu5s · 2025-10-29

**Soundness:** 2
**Presentation:** 2
**Contribution:** 2
**Rating:** 2
**Confidence:** 3

**Summary:**

This paper introduces a framework for interpreting classifier weights by decomposing them into "principal components" (i.e. associated to non-zero singular values) and "null components" (i.e. the remaining ones). In parallel, the method learns a mapping to project classifier features into the CLIP embedding space for enhanced interpretability. The core idea involves removing from the classifier features their null components, and then translating them into the CLIP space. The authors then present several analyses to interpret these translated features, including measuring changes in similarity with various concepts and visualizing features by steering along directions defined by the null components.

**Strengths:**

- **Originality:** The proposed decomposition of classifier weights into principal and null components is an original perspective for interpretability. This approach can be useful to analyze the functional aspects of learned representations.
- **Clarity of method:** The method overview figure is very helpful to understand the overall flow of the proposed framework.

**Weaknesses:**

- **Clarity of Experiments and Findings:** The description of the experimental setup and the interpretation of the results lack clarity. It is difficult to understand what specific questions each experiment aims to answer and what conclusions should be drawn from the presented findings. More detailed explanations and clearer connections between experimental results and their implications would greatly improve this section.
- **Quality:** The paper does not directly verify that the decomposition and CLIP mapping preserve the original classifier’s logits/predictions. While the decomposition itself, by construction, should ideally preserve the logit outputs, the framework makes a lot of additional approximations throughout the pipeline.
- **Significance:** The practical applications and broader significance of this work are not clear. While the paper mentions potential sensitivity to semantic shifts, spurious correlations, and debugging failure cases, it does not clearly articulate _how_ the proposed framework can be used to address these issues, and does not provide examples in practice.

**Questions:**

- How can this method be applied to fix or improve mistakes made by models?
- What specific types of spurious correlations could be identified using this approach?
- Could you provide concrete examples or case studies demonstrating the utility of the framework and the analysis?
- Could you provide an empirical verification that the logits and predictions remain unchanged for the "equivalent features"?

---

### Official Review · Reviewer_bV6R · 2025-11-04

**Soundness:** 2
**Presentation:** 3
**Contribution:** 2
**Rating:** 2
**Confidence:** 4

**Summary:**

This paper introduces a method called SING (Semantic Interpretation of the Null-space Geometry) that analyzes the semantic invariants in image classifiers. The core idea is that all architectures have invariants (sets of inputs that produce identical outputs), which reside in the nullspace of the classifier, and are hard to interpret semantically. Authors propose a method that can be used to get natural language descriptions and visual examples of the semantic shifts that happen in the null space. My initial recommendation is reject, even though this work is a very good attempt and has potential to be impactful if several key missing aspects are addressed.

**Strengths:**

The method can be applied to a single image to uncover local invariants, as well as sets of images for statistical analysis.

The method reveals important insights about different model architectures. E.g., authors show that ResNet50 leaks relevant semantic attributes to the null space, while DINO-ViT is better at maintaining class semantics across the invariant space.

The paper demonstrates how this approach can help understand the robustness of different neural network architectures, diagnose problematic semantic invariants, and reveals what types of information classifiers ignore or maintain when making predictions.

The motivation of the paper is well-founded. As the use of AI systems becomes more widespread and critical our society, it is increasingly important that we understand the fundamental properties of classification systems. Thus, we would do well to diagnose what information such models are actually using vs. ignoring. This can provide valuable insights, and hopefully can lead to more reliable and trustworthy systems.

**Weaknesses:**

I am fascinated by the fact that Dino shows better results than other architectures. Why is that the case? What does that tell us about Dino? What does that tell us about SING as a framework? Unfortunately, the authors do not provide any type of explanation as to why that is the case. While it is helpful to have SING as a framework, if we do not understand the reason behind its results, it is difficult to truly understand and trust it.

The paper assumes a linear approximation. I wonder why this is the case. Did the authors opt for the simplest solution? This can often be the better way to go -- the simpler, the better. However, wouldn't this linear approximation miss more complicated invariant structures?

The framework heavily relies on CLIP. How will CLIP's limitations affect this work?

A simple low-hanging fruit is ablation studies that would be valuable to the work.

An important missing one is comparison to baselines / interpretablity methods.

It would be excellent to see user studies. I.e., how are the results produced by SING useful to people in practice?

What is the computational complexity of the method? How does it scale?

The theoretical framework is not rigorous. There is plenty of work in the literature where authors could draw from to make a stronger case. Invariants, nullspaces, etc, are widely studied. Authors can draw information from high-dimensional geometry (e.g., the the manifold hypothesis, Riemannian optimization literature) to provide a stronger theoretical foundation -- or at least an argument -- for why the work is rigorous and should exist.

The work is a very good start, an interesting result, and can potentially be extremely helpful to the community. It certainly required proficiency and creativity, and I believe with some improvements and perhaps another iteration at another venue this work could be very strong.

**Questions:**

In the weaknesses section, I have specified many questions.

---

### Note · Authors · 2025-11-12

**Comment:**

We have read the reviewers inputs and decided to create an improved version for another iteration

**Withdrawal Confirmation:**

I have read and agree with the venue's withdrawal policy on behalf of myself and my co-authors.